# Association of the Naples Prognostic Score with Long-Term Adverse Events in Chronic Limb-Threatening Ischemia After Below-the-Knee Endovascular Revascularization

**DOI:** 10.3390/diagnostics14232627

**Published:** 2024-11-22

**Authors:** Emir Dervis, Aykun Hakgor, Muhammed Mert Goksu, Idris Yakut, Hasan Can Konte, Cafer Panc, Ismail Gurbak, Ali Kemal Kalkan, Hamdi Pusuroglu, Ahmet Arif Yalcin, Mehmet Erturk

**Affiliations:** 1Department of Cardiology, Medipol University, Istanbul 34810, Türkiye; aykunhakgor@gmail.com (A.H.); idrislive@windowslive.com (I.Y.); hcankonte@gmail.com (H.C.K.); 2Department of Cardiology, Basaksehir Cam and Sakura City Hospital, Istanbul 34480, Türkiye; dr.mmgoksu@gmail.com (M.M.G.); hpusts@gmail.com (H.P.); 3Department of Cardiology, University of Health Sciences, Istanbul Mehmet Akif Ersoy Thoracic and Cardiovascular Surgery Training and Research Hospital, Istanbul 34303, Türkiye; caferpanc@gmail.com (C.P.); drakkalkan@gmail.com (A.K.K.); yalcin_arif@hotmail.com (A.A.Y.); drerturk@gmail.com (M.E.); 4Department of Cardiology, Istinye University, Istanbul 34396, Türkiye; ismailgurbak@gmail.com

**Keywords:** peripheral arterial disease, chronic limb-threatening ischemia, Naples prognostic score, mortality, adverse events

## Abstract

Objectives: Chronic limb-threatening ischemia (CLTI) is the most severe manifestation of peripheral artery disease (PAD) and is associated with high morbidity and mortality. The Naples prognostic score (NPS), a composite marker incorporating serum albumin, total cholesterol, neutrophil-to-lymphocyte ratio (NLR), and lymphocyte-to-monocyte ratio (LMR), has shown prognostic value in various cardiovascular conditions. This study aimed to evaluate the prognostic significance of the NPS in predicting all-cause mortality and any kind of amputation in patients with CLTI undergoing endovascular treatment (EVT) for below-the-knee (BTK) lesions. Methods: In this retrospective analysis, 191 patients diagnosed with CLTI and treated with EVT for BTK lesions between 2017 and 2023 were stratified into three groups based on the NPS: low (0–1), intermediate (2), and high (3–4). The primary endpoint was all-cause mortality, while the secondary endpoint was any kind of amputation. Results: A higher NPS was significantly associated with increased all-cause mortality (hazard ratio: 3.66; 95% confidence interval: 1.72–7.78; *p* < 0.001), while no significant association was observed between the NPS and major amputation. Independent predictors of mortality included a high NPS, reduced left ventricular ejection fraction, and impaired renal function. Conclusions: The NPS is an independent predictor of long-term mortality in CLTI patients undergoing EVT for BTK lesions.

## 1. Introduction

Chronic limb-threatening ischemia (CLTI) represents the most severe manifestation of peripheral artery disease (PAD), often leading to significant morbidity and mortality. It is characterized by chronic ischemic rest pain, non-healing wounds, or gangrene in one or both legs, which substantially impairs quality of life. CLTI is associated with a high risk of major adverse cardiovascular events and limb loss, making effective management crucial [1,2]. Endovascular treatment (EVT) is a less invasive alternative to surgical revascularization, which is primarily performed for short lesions or in patients with high surgical risk or in patients for whom a suitable autologous venous graft is not available [3,4]. According to the studies, the mortality rate remains high, possibly due to concomitant comorbidities, in patients with CLTI even if EVT is performed [5].

In this context, the Naples prognostic score (NPS), a composite index derived from nutritional and inflammatory markers, has been introduced as a potential predictor of clinical outcomes in cancer patients [6,7,8]. This scoring system includes parameters such as serum albumin, total cholesterol, neutrophil-to-lymphocyte ratio (NLR), and lymphocyte-to-monocyte ratio (LMR), which together provide insights into the patient’s nutritional and inflammatory status [6]. In addition to the prognosis of cancer patients, the NPS has recently been shown to predict prognoses in cardiovascular diseases, such as heart failure and valvular diseases [9,10]. However, its prognostic value in the specific setting of CLTI treated with below-the-knee (BTK) endovascular procedures remains unclear. Understanding this relationship is essential, as it could enhance risk stratification and guide therapeutic decisions, ultimately improving patient outcomes.

This study aims to elucidate the relationship between the NPS and long-term adverse events, including all-cause mortality and any type of amputation, in patients with CLTI undergoing EVT for BTK arteries, thereby enhancing our understanding of its utility in this high-risk population.

## 2. Materials and Methods

### 2.1. Study Population and Design

This retrospective study included 191 patients with CLTI who underwent EVT for BTK lesions in tertiary care centers (Mehmet Akif Ersoy Thoracic and Cardiovascular Surgery Training and Research Hospital, Basaksehir Cam and Sakura City Hospital) between 2017 and 2023. The exclusion criteria for our study were as follows: an absence of laboratory parameters, active infection requiring antibiotic therapy, advanced hepatic (Child–Pugh class C) failure, malignancy, or previous intervention (percutaneous or surgical approach) to the same vascular bed. When the databases of the hospitals were searched, a total of 782 patients who underwent EVT for BTK procedures were found and 191 patients remained after the exclusion criteria were applied. Information on demographic data, laboratory findings, and morbidity and mortality data were collected from electronic medical records and telephone interviews. This study adhered to the principles of the Helsinki Declaration and received approval from the Ethics Committee of Istanbul Medipol University. Due to the retrospective and observational nature of this study, written informed consents could not be obtained from the patients.

### 2.2. Endovascular Therapy and Follow-Up

Procedures were performed under local anesthesia by an experienced interventional cardiologist. Loading doses of aspirin (100–300 mg) and clopidogrel (300–600 mg) were given prior to the index procedure. Vascular access was typically obtained through the femoral artery and usually the antegrad ipsilateral approach was the preferred option. Following the establishment of arterial access, patients received an initial bolus dose of unfractionated heparin (70–100 units/kg). Based on lesion characteristics, EVT procedures were performed. Following revascularization, angiographic success was described as achieving direct blood flow with a residual stenosis of 40% or less. After the procedure, patients have taken dual antiplatelet therapy with 75–100 mg acetylsalicylic acid (ASA) and 75 mg clopidogrel for 1–3 months according to their clinical characteristics. After that, monotherapy (ASA or clopidogrel) was continued. Depending on the other comorbidities of the patients, concomitant medical treatments, such as statins, antihypertensive medications, and beta blockers, were adjusted according to the latest guidelines. Follow-up assessments were conducted at 1, 3, 6, and 12 months post-procedure and every year thereafter. In outpatient clinic follow-ups, a clinical evaluation, ankle–brachial index (ABI) measurement, and duplex ultrasonography were performed to assess patency.

### 2.3. Endpoints of This Study

This study’ s primary endpoint was long-term all-cause mortality. Mortality data have been confirmed from the national database system. Major or minor amputation was this study’s secondary endpoint. Major amputation was defined as any amputation performed at a level above the ankle.

### 2.4. Laboratory Analysis

Blood samples, including the parameters analyzed for NPS calculation, were collected from all patients after an eight-hour fasting state before the procedure. As described in a recent article, the NPS was calculated using four parameters: serum albumin concentration, total cholesterol (TC) concentration, NLR, and LMR. Specifically, a serum albumin concentration of less than 4 g/dL was given 1 point, while a concentration of 4 g/dL or higher was given 0 points. For TC concentration, levels below 180 mg/dL were assigned 1 point, and levels of 180 mg/dL or higher were assigned 0 points. An NLR of 2.96 or higher was scored as 1 point, whereas an NLR below 2.96 was scored as 0 points. An LMR below 4.44 was assigned 1 point, while an LMR of 4.44 or higher was assigned 0 points. The NPS was the total sum of these individual scores. In this study, patients were categorized into three groups based on their NPS: those with a score of 0 or 1 were placed in the low-NPS group; those with a score of 2 were assigned to the intermediate-NPS group; and those with a score of 3 or 4 were classified as the high-NPS group (Figure 1).

### 2.5. Statistical Analysis

All statistical analyses were performed using IBM SPSS for Windows version 26.0 (SPSS, Chicago, IL, USA). Kolmogorov–Smirnov tests were performed to assess the normality of the data distribution. Continuous variables were expressed as the mean± standard deviation and median (25th–75th percentiles), and categorical variables were expressed as counts (percentage). Comparisons of continuous variables between the groups were performed using the ANOVA and Kruskal–Wallis tests. Comparisons of categorical variables between the groups were performed using the Yates’ and Monte Carlo chi-squared test. The cohort was categorized into three different groups based on the NPS, as outlined in Table 1. To determine the predictors of all-cause mortality and composite endpoint, including all-cause mortality and any kind of amputation, Cox regression analysis was performed. Independent predictors of all-cause mortality and their adjusted hazard ratios (HRs) and 95% confidence interval (CI) were evaluated using a multivariate model created by including variables that achieved statistical significance (*p* < 0.05) in the univariate analysis. Survival rates were plotted using the Kaplan–Meier method. A two-sided *p*-value < 0.05 was considered statistically significant.

## 3. Results

### 3.1. Comparison of Baseline Characteristics

This study includes a total of 191 patients with CLTI who underwent EVT for BTK lesions. Patients were divided into three groups based on their NPS. In the baseline characteristics, it was observed that patients were younger in the low-NPS group compared with the other groups (*p* = 0.002). Additionally, the number of patients with diabetes mellitus (DM) (*p* = 0.035) or chronic kidney disease (CKD) was higher in the high-NPS group. The number of patients with CKD (*p* = 0.003) was also greater in the high-NPS group compared to the other groups. Moreover, the prevalence of heart failure (HF) was higher in the high-NPS group (*p* = 0.006), and these patients had lower left ventricular ejection fraction (LVEF) values (*p* = 0.004). No significant differences were found in other baseline characteristics between the groups.

No statistically significant differences were observed in other BTK artery involvements between NPS groups. In terms of medication use patterns, no significant differences were found between the groups.

In blood test analyses, a decline in hemoglobin levels was observed as the NPS increased (*p* = 0.001). There was also a significant decline in glomerular filtration rate (GFR) as the NPS increased (*p* < 0.001) (Table 1).

### 3.2. Adverse Events Across NPS Groups

All-cause mortality rates were 3 (10.3%), 7 (11.9%), and 47 (45.6%) in the low-, intermediate-, and high-NPS groups, respectively, within a median follow-up period of 26.1 (17.5–39.6) months (*p* < 0.001). No significant statistical differences were found between the groups in terms of major, minor, or any amputation rates (*p* > 0.05 for all). Accordingly, the rates of composite outcome, defined as any kind of amputation or all-cause mortality at long-term follow-up, were 7 (24.1%), 16 (27.1%), and 67 (65%) in the low-, intermediate-, and high-NPS groups, respectively (*p* < 0.001) (Table 2).

### 3.3. Determinants of Long-Term Survival

According to univariable Cox regression analysis, age, GFR, LVEF, current smoking, Rutherford class 4–6, and being in the high-NPS group were associated with long-term all-cause mortality in patients undergoing EVT for PAD (Table 3).

According to the multivariable model, having certain LVEF covariates [adj OR: 0.973, 95% CI: 0.949–0.999, *p* = 0.038] and being in the high-NPS group compared to the low-NPS [adjOR: 3.660, 95% CI: 1.075–12.458, *p* = 0.038] were found to be independent predictors of long-term mortality (Table 3).

According to Kaplan–Meier survival curves, an increased mortality rate that did not reach statistical significance was observed in the intermediate-NPS group compared to the low-NPS group, while the high-NPS group showed a 6.1-fold increased all-cause mortality rate compared to the low-NPS group (Figure 1).

### 3.4. Major and Minor Amputation

Based on the model that included Rutherford class 4–6, PTA disease, DM, hyperlipidemia (HL), and being under ASA treatment, the independent predictors for any type of amputation were Rutherford class 4–6 [adjOR: 4.176, 95% CI: 1.622–10.752, *p* = 0.003], PTA disease [adjOR: 2.614, 95% CI: 1.410–4.846, *p* = 0.002], HL [adjOR: 2.001, 95% CI: 1.052–3.806, *p* = 0.035], and being under ASA treatment [adjOR: 0.370, 95% CI: 0.195–0.704, *p* = 0.002] (Table 4).

### 3.5. Abbreviations: OR: Odds Ratio, PTA: Posterior Tibial Artery

#### Composite Endpoint

The association of the study population variables with the composite endpoint of long-term all-cause mortality or any kind of amputation is given in Table 5.

Rutherford class 4–6 [adjOR: 2.520, 95% CI: 1.463–4.341, *p* = 0.001], PTA disease [adjOR: 1.541, 95% CI: 1.003–2.367, *p* = 0.049], LVEF [adjOR: 0.979, 95CI: 0.958–1.000, *p* = 0.045], and the high-NPS group compared to the low-NPS group [adjOR: 2.458, 95% CI: 1.051–5.745, *p* = 0.038] were found to be independent predictors of composite outcomes (Table 5). However, these results are primarily driven by mortality outcomes, as the statistical analysis did not show a significant association between the NPS and the amputation rates. For the composite outcomes, Kaplan–Meier analysis demonstrated a 3.58-fold increased risk in the high-NPS group compared to the low-NPS group (Figure 2). 

## 4. Discussion

In the current study, we investigated the relationship between the NPS and long-term mortality and amputation in patients with CLTI who have BTK lesions and were revascularized with EVT. Our data demonstrated that a higher NPS was independently associated with all-cause mortality; however, no relationship was found between amputation rates and the NPS. In addition, the composite outcome was created to broaden the clinical application of the NPS by capturing multiple adverse events that are clinically relevant in this high-risk patient population. It has been determined that a higher NPS is associated with the composite endpoint that includes both amputation and all-cause mortality. This association is believed to be largely attributable to the strong statistical correlation between the NPS and mortality. In addition to a high NPS, a low LVEF was identified as an independent predictor of mortality.

Despite increasing revascularization rates and improvements in medical treatment, high mortality rates persist in PAD [4,5]. In patients with symptomatic PAD, the 5-year incidence of CV mortality is 13%. However, compared to symptomatic PAD, CLTI significantly raises the risk of both mortality (RR 2.26) and major adverse cardiovascular events (MACEs) (RR 1.73) [11]. In patients with CLTI, it is recommended that, for limb salvage, revascularization should be performed. The management of BTK arterial occlusive disease remains a challenging condition, due to both the patients’ overall clinical status and the complexities of revascularization strategies. EVT is recommended as the first-line approach in patients with CLTI with BTK lesions who present with short lesions or in those with elevated surgical risk or lacking a suitable autologous saphenous vein graft for bypass [3,4]. Following revascularization with EVT, restenosis rates in BTK lesions exceeded 50%. This study also demonstrated that the 5-year survival rate was 48% [12]. Identifying factors associated with mortality and MACEs in this high-risk patient population is crucial for optimizing management strategies.

Atherosclerosis is a complex condition influenced by multiple factors, where inflammatory markers play significant roles in both treatment and prognosis. PAD is an imminent manifestation of atherosclerotic cardiovascular disease and its prevalence has increased in recent years despite the widespread adoption of primary prevention strategies [13]. Inflammation is one of the major factors in the beginning and advancement of PAD [14]. For instance, some inflammatory cytokines, such as interleukin-6 and tumor necrosis factor-α (TNF-α), have been shown to be closely associated with the incidence of PAD [15]. In addition, compared to expensive and infrequently used tests, the complete blood count (CBC) is an easily accessible, relatively inexpensive, and commonly used laboratory parameter that provides information about the inflammatory process. NLR, a readily calculable parameter derived from the CBC, offers significant insights into inflammation. In the study conducted by Aykan et al., it was shown that the NLR may have an impact on PAD complexity [16]. Also, it has been demonstrated in a study that an elevated NLR was independently associated with severe multi-level PAD [17]. In another study, an elevated NLR was found to be associated with increased mortality and amputation rates during the follow-up of patients who underwent EVT [18]. The LMR, another inflammatory marker derived from the CBC, has also been previously identified as being associated with atherosclerosis in earlier studies. Furthermore, it has been demonstrated that a low LMR is associated with the development of CLTI in patients with PAD [19]. In addition, Cosarca et al. have shown that reduced LMR levels were associated with higher amputation rates after revascularization [20].

Some clinical studies have also shown that malnutrition, assessed through various biomarkers in different patient profiles, is associated with increased cardiovascular mortality [21,22]. In a recently published study evaluating adverse events following revascularization in patients with PAD, malnutrition was found to be associated with major adverse events after revascularization [23]. Furthermore, a review of studies on patients with CLTI reported that low serum albumin levels, used as a marker of malnutrition, were related to increased mortality [24]. In another study it has been shown that moderate-to-severe malnutrition, as defined by the Controlling Nutritional Status (CONUT) score, was found to be significantly associated with increased mortality in patients with CLTI after undergoing EVT.

The NPS is a recent marker that provides information on both the nutritional and inflammatory status of patients. It incorporates serum albumin, TC, NLR, and LMR. The NPS was first introduced in the literature in 2017 by Galizia et al. as a predictor of long-term prognosis following surgery in patients with colorectal cancer [6]. Recent studies have also demonstrated that, beyond malignancy, the NPS is an independent predictor of prognosis in atherosclerotic diseases [9,10,25]. In our study, a high NPS was shown to be a predictor of all-cause mortality in the follow-up of patients who underwent EVT for BTK lesions. Our analysis revealed no significant difference in all-cause mortality between the groups with a low and medium NPS. Only a high NPS was shown to be significantly associated with all-cause mortality, with a 3.66-fold increase in mortality compared to the low-NPS group [95% CI (1.075–12.458), *p* = 0.038].

Considering the impact of the NPS, an indicator of inflammatory and nutritional status, on mortality following EVT, addressing nutritional and inflammatory processes in this vulnerable patient group is of critical importance. A recent study demonstrated that negative acute-phase reactants increased while positive acute-phase reactants decreased in patients with CLTI three months after revascularization [14]. According to the results of this study, in order to partially reduce the inflammatory process, complete revascularization of all suitable vessels may be considered, particularly in patients with a high NPS. In addition, for patients with foot ulcers, proper wound care is crucial to mitigate chronic inflammation beyond revascularization. Furthermore, addressing malnutrition through nutritional support, especially in the high-NPS group, could be considered to improve both the patients’ quality of life and their nutritional status.

Though a high NPS seems to be useful in predicting the prognosis of patients with PAH, it raises some questions. The use of lipid-lowering therapies is strongly recommended for high-risk atherosclerotic patients. Recent guidelines have set a target LDL level of less than 55 mg/dL for patients with PAD [4]. This situation brings questions about the reliability of the NPS in this patient population, as lipid-lowering treatments will naturally decrease TC values. However, since the NPS encompasses four parameters, reflecting not only TC levels but also the patient’s inflammatory status and nutritional state, we believe that lipid-lowering therapy alone does not compromise the NPS’s predictive accuracy for survival. Nonetheless, developing a modified NPS that adjusts total cholesterol values in patients with atherosclerosis receiving lipid-lowering therapy could be valuable, warranting further research based on this adapted scoring system.

In our study, multivariate logistic regression analysis identified a low LVEF, in addition to the high NPS, as an independent predictor of all-cause mortality. Previous studies have also demonstrated that a low LVEF is a predictor of mortality in patients with PAD [5,26]. The results of our current study appear to be consistent with findings reported in the literature.

In addition to mortality, amputation was defined as the secondary endpoint in our study. It was shown that a high NPS did not have an effect on predicting amputation. According to multivariate regression analyses, not being on ASA therapy, having a Rutherford score between 4 and 6, the presence of a lesion in the PTA site, and HL were identified as predictors of amputation.

For the composite endpoint, which includes either mortality or amputation, a high NPS was found to have an impact on the outcome. Although we acknowledge that composite outcome results are mostly associated with mortality in our study, we believe that the NPS could enhance its prognostic utility by providing a more comprehensive assessment of patient risk through a composite outcome that includes both mortality and amputation events. With the statistical significance of composite outcome, it can be concluded that the NPS is generally associated with poor outcomes on the basis of amputation or mortality. In addition to the NPS, a low LVEF and a Rutherford classification of 4–6 were also predictors of the composite outcome.

Although our study focused on EVT for BTK arteries, patients with CLTI could present with multivessel disease [27]. Consequently, some individuals in the study cohort also underwent EVT for proximal arteries, aiming to achieve optimal perfusion and improved outcomes. However, statistical analyses revealed no significant differences between groups with and without proximal lesions in terms of mortality, amputation rates, or between NPS degrees.

Based on the results of our study, the NPS was shown to be an independent predictor of long-term mortality and the composite endpoint of mortality plus amputation in patients with CLTI who underwent EVT for BTK lesions. Although this scoring system is not currently included in clinical guidelines, it could potentially serve as a useful tool for identifying long-term risks in this patient group.

### Study Limitation

This study has some limitations. The first limitation of this study is the relatively small sample size and its retrospective nature. Second, the NPS was calculated only prior to the procedure, and changes in the NPS during follow-up might have occurred, which may affect the patient’s clinical status. Understanding how the NPS fluctuates over the long-term follow-up could provide unique insights and further differentiate this study’s findings. Third, we did not investigate the correlation of other inflammatory markers and objective nutritional indices with the NPS.

## 5. Conclusions

In the long-term follow-up of patients with CLTI who underwent EVT for BTK lesions, the NPS was found to be associated with mortality. As a simple and inexpensive biomarker, the NPS appears to be a useful tool for predicting the prognosis of these patients. It could be employed for the close monitoring, treatment, and management of this high-risk and fragile patient group. However, larger-scale and prospective studies are needed to further validate its reliability and effectiveness.

## Data Availability

The data supporting this study are available from the corresponding author upon reasonable request.

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
