# Peer review of "Association of the Naples Prognostic Score with Long-Term Adverse Events in Chronic Limb-Threatening Ischemia After Below-the-Knee Endovascular Revascularization"

_diagnostics, 2024, doi:10.3390/diagnostics14232627_

Round 1

Reviewer 1 Report

Comments and Suggestions for Authors

Interesting contribution in the detection of fragile CLTI patients undergoing BTk EVT. Following are my suggestions to improve the manuscript.

Author List: authors’ names must be reported according to Diagnostics’ Instructions for Authors (Front Matter).

Abstract

- line 17, “Chronic limb-threatening ischemia (CLI) …”: the correct (world-wide recognized) acronym is CLTI;

- line 22, “… EVT … BTK …”: any acronym must be explained the first time it is introduced;

- lines 27 and 29, “… HR … CI … LVEF …”: there is no need to introduce these acronyms, since you cite each of them just once in the Abstract. Save space.

 Introduction 

- line 35, “Chronic critical limb ischaemia (CLI) …”: be consistent with the best terminology, use “Chronic limb-threatening ischemia (CLTI) …”, as in the Abstract.

Materials and Methods

- Endovascular Therapy and Follow-up, line 85, “… ASA …”: any acronym must be explained the first time it is introduced; furthermore, correct klopidogrel in Clopidogrel.

- Endpoints of the study, line 96-7, “We defined also another secondary endpoint consisting of all-cause mortality …”: I guess this is a typo. All-cause mortality is the primary endpoint of your study;

- Laboratory Analysis, line 102, “… TC concentration …”: any acronym must be explained the first time it is introduced.

Statistical Analysis

- line 126, “… HRs … Cis …”: any acronym must be explained the first time it is introduced.

Results 

- Comparison of baseline characteristics, line 134, “… DM …”: any acronym must be explained the first time it is introduced.

- Table 2, 3, and 4 should contain at their end an Abbreviations section explaining the meaning of NPS, OR, and PTA.

- Table 3: GFR has a p value = 0.048 which, when approximated, is =0.05, not <0.05. Please, consult a professional statistitian. Part of your results might change.

- Major and minor amputation, line 192, “… HL …”: any acronym must be explained the first time it is introduced.

Discussion

- lines 229-30, “Despite increasing revascularization rates and improvements in medical treatment, high mortality rates persist in PAD.”: you need an appropriate and recent reference, such as: (Martelli, E.; Enea, I.; Zamboni, M.; Federici, M.; Bracale, U.M.; Sangiorgi, G.; Martelli, A.R.; Messina, T.; Settembrini, A.M. Focus on the Most Common Paucisymptomatic Vasculopathic Population, from Diagnosis to Secondary Prevention of Complications. Diagnostics (Basel) 202313, 2356-2376);

- lines 254-5, “… NLR may have an impact on peripheral artery disease complexity …”: be consistent, use the acronym PAD.

References # 7, 21, 22, and 24 should be reported  according to Diagnostics’ Instructions for Author (see the suggested reference above, as an example). 

Author Response

Thank you very much for your thorough review of our manuscript and for your contributions. We appreciate your positive assessment of our study and have carefully considered your suggestions to further enhance the manuscript. Below, we outline our responses to each point:

Comments 1 : Author List: authors’ names must be reported according to Diagnostics’ Instructions for Authors (Front Matter).

Response 1: We have adjusted the format of the author names to comply with the Diagnostics Instructions for Authors.

Comments 2: Abstract

- line 17, “Chronic limb-threatening ischemia (CLI) …”: the correct (world-wide recognized) acronym is CLTI;

- line 22, “… EVT … BTK …”: any acronym must be explained the first time it is introduced;

- lines 27 and 29, “… HR … CI … LVEF …”: there is no need to introduce these acronyms, since you cite each of them just once in the Abstract. Save space.

Response 2: Abstract:

  • Line 17: We have corrected “CLI” to “CLTI” to align with the globally accepted terminology.
  • Line 22: We have explained all acronyms (e.g., EVT, BTK) at their first occurrence for clarity.
  • Lines 27 and 29: We removed the acronyms as your suggestion.

Comments 3 : Introduction 

- line 35, “Chronic critical limb ischaemia (CLI) …”: be consistent with the best terminology, use “Chronic limb-threatening ischemia (CLTI) …”, as in the Abstract.

Response 3: Introduction:

  • Line 35: We have replaced “[1](CLI)” with “Chronic limb-threatening ischemia (CLTI)” as your suggestion

Comments 4: Materials and Methods

- Endovascular Therapy and Follow-up, line 85, “… ASA …”: any acronym must be explained the first time it is introduced; furthermore, correct klopidogrel in Clopidogrel.

- Endpoints of the study, line 96-7, “We defined also another secondary endpoint consisting of all-cause mortality …”: I guess this is a typo. All-cause mortality is the primary endpoint of your study;

- Laboratory Analysis, line 102, “… TC concentration …”: any acronym must be explained the first time it is introduced.

Response 4: Materials and Methods:

  • Endovascular Therapy and Follow-up, Line 85: We have provided the expansion of “ASA” as acetylsalicylic acid at its first appearance and corrected “klopidogrel” to “clopidogrel.”
  • Endpoints of the study, Lines 96-97: Thank you for pointing this out. We have corrected the description to accurately reflect that all-cause mortality is the primary endpoint of our study. We have removed the sentence that you cited.
  • Laboratory Analysis, Line 102: We have explained “TC” at its first mention.

Comments 5: Statistical Analysis

- line 126, “… HRs … Cis …”: any acronym must be explained the first time it is introduced.

Response 5: Statistical Analysis:

  • Line 126: We have expanded the acronyms (“HRs” and “CIs”)

Comments 6: Results 

- Comparison of baseline characteristics, line 134, “… DM …”: any acronym must be explained the first time it is introduced.

- Table 2, 3, and 4 should contain at their end an Abbreviations section explaining the meaning of NPS, OR, and PTA.

- Table 3: GFR has a p value = 0.048 which, when approximated, is =0.05, not <0.05. Please, consult a professional statistitian. Part of your results might change.

- Major and minor amputation, line 192, “… HL …”: any acronym must be explained the first time it is introduced.

Response 6: Results:

  • Comparison of Baseline Characteristics, Line 134: We have defined “DM” at its first occurrence.
  • Tables 2, 3, and 4: We have added the meanings of “NPS,” “OR,” and “PTA” at the end of each table.
  • Table 3: Thank you for your valuable suggestion and information. As you said, with approximation p-value is 0.05. Beside the CI was found between 0.981-1.000 which reflects that GFR is not strongly significant predictor on all-cause mortality. Therefore we made the corrections which are about GFR and its prediction of mortality, in results and discussion section.
  • Major and Minor Amputation, Line 192: We have expanded “HL” at its first mention.

Comments 7: Discussion

- lines 229-30, “Despite increasing revascularization rates and improvements in medical treatment, high mortality rates persist in PAD.”: you need an appropriate and recent reference, such as: (Martelli, E.; Enea, I.; Zamboni, M.; Federici, M.; Bracale, U.M.; Sangiorgi, G.; Martelli, A.R.; Messina, T.; Settembrini, A.M. Focus on the Most Common Paucisymptomatic Vasculopathic Population, from Diagnosis to Secondary Prevention of Complications. Diagnostics (Basel) 202313, 2356-2376);

- lines 254-5, “… NLR may have an impact on peripheral artery disease complexity …”: be consistent, use the acronym PAD.

Response 7: Discussion:

  • Lines 229-230: We have added the recent reference on high mortality rates in PAD as your suggestion
  • Lines 254-255: We have made the correction “peripheral artery disease” to “PAD”

Comments 8: References # 7, 21, 22, and 24 should be reported  according to Diagnostics’ Instructions for Author (see the suggested reference above, as an example). 

Response 8: References:

  • We have revised References #7, #21, #22, and #24 to align with the Diagnostics formatting guidelines.

Thank you once again for your constructive feedback and we hope the revised version meets the expectations of the journal.

Reviewer 2 Report

Comments and Suggestions for Authors

The authors present an original study entitled “Association of the Naples Prognostic Score with Long-Term Adverse Events in Chronic Limb-Threatening Ischemia After Below-the-Knee Endovascular Revascularization”.

The results are quite interesting and have practical relevance. The manuscript has a good scientific soundness. The methodology is described in some detail.

I have a few points to address:

1.    The authors use different terms in the abstract and then in the text of the manuscript – “Chronic limb-threatening ischemia (CLI)” and “Chronic critical limb ischaemia (CLI)”. The authors should choose one term, probably CLTI.

2.    The PAD patient population represents patients at very high cardiovascular risk who should receive hypolipidemic therapy and achieve target LDL-С levels of less than 55 mg/dl or 1.4 mmol/L. In the sample presented by the authors, the frequency of administration of statins was quite low, which is generally consistent with the literature. However, how appropriate in this population is it to use a score that considers low cholesterol values, which should be achieved in this population, as an additional risk factor. Is it possible that counting this item on the NPS scale requires taking into account whether or not the patient is taking statins? The authors should briefly address this in the discussion. Perhaps a modification of the NPS scale is needed for patients with atherosclerotic CVD?

Author Response

Thank you for your valuable feedback and insightful comments regarding our manuscript, “Association of the Naples Prognostic Score with Long-Term Adverse Events in Chronic Limb-Threatening Ischemia After Below-the-Knee Endovascular Revascularization.” We appreciate your positive assessment of our study and have carefully considered your suggestions to further enhance the manuscript. Below, we outline our responses to each point:

Comments 1: The authors use different terms in the abstract and then in the text of the manuscript – “Chronic limb-threatening ischemia (CLI)” and “Chronic critical limb ischaemia (CLI)”. The authors should choose one term, probably CLTI.

Response 1: Dear reviewer, thank you your recommendation to standardize the terminology. We have revised the manuscript to consistently use the term “Chronic limb-threatening ischemia (CLTI)” throughout the Abstract and main text to align with current, globally recognized terminology.

Comments 2: The PAD patient population represents patients at very high cardiovascular risk who should receive hypolipidemic therapy and achieve target LDL-С levels of less than 55 mg/dl or 1.4 mmol/L. In the sample presented by the authors, the frequency of administration of statins was quite low, which is generally consistent with the literature. However, how appropriate in this population is it to use a score that considers low cholesterol values, which should be achieved in this population, as an additional risk factor. Is it possible that counting this item on the NPS scale requires taking into account whether or not the patient is taking statins? The authors should briefly address this in the discussion. Perhaps a modification of the NPS scale is needed for patients with atherosclerotic CVD?

Response 2: Dear reviewer, we appreciate your insightful comment. First of all, when we analysed Methods section of the studies, which are about the effect of NPS on outcomes, statin usage is not the exclusion criteria of these studies. Therefore, in our study we have decided to include the patients under statin treatment. However, as you mentioned, the targeted LDL-C levels are below 55 mg/dl in patients with peripheral artery disease, thus in this patient group NPS score could be expected false higher levels due to the lower TC levels. In this patient group, it could be seen as a disadvantage of NPS scale. However, NPS scale is not consist of only one parameter which reflects patient’ s nutritional status. It contains extra 3 additional paramters which reflect patient’ s nutritional and inflammatory status. Therefore, we believe in NPS can show patient’ s general status, independenly of statin treatment. However, we have shared your thoughts and we recognize that this factor could warrant consideration in future modifications of the NPS scale to account for statin use as a confounding factor when assessing patients with established ASCVD. We also acknowledge the need for further studies to investigate how risk scores like NPS might be adapted or recalibrated for ASCVD populations undergoing optimal lipid-lowering therapy. Therefore, as your suggestion, we have revised the manuscript (line 297-306).

Thank you once again for your constructive feedback and we hope the revised version meets the expectations of the journal.

Reviewer 3 Report

Comments and Suggestions for Authors

Authors have investigated an interesting topic, possible association between the Naples Prognostic Score (NPS) and outcome after endovascular treatment of below the knee (BTK) arteries. NPS is a composite index derived from nutritional and inflammatory markers, has been introduced as a potential predictor of clinical outcome on patients with several cancers.

As for the treatment of BTK arteries patient selection should be focused on CLI/CLTI. The nomenclature should be revised for the manuscript since CLI originates from TASC II and would include either ABI or TBI etc with below values stated in the TASC II guideline. CLTI might be more appropriate instead of CLI since no measurements above mentioned values are presented or sTO2.

Although some critics the manuscript is well written and would need only some revision.

Abstract: write out EVT and BTK. CLI right terminology?

Introduction: Only comment the above mentioned CLI/CLTI would need attention

Methods: Clopidogrel loading dose up to 600 mg? Antitrombotic treatment after first 1 month missing or not in use? ABI measured after the procedure. Not before?

Results: Embolectomy table 1? Embolectomy referes to an open procedure? No ABI values presented, please see general comment 

Line 191 minor instead of minör

Composite outcome seem to be mortality driven since the major and minor amputations are not associated with NPS.

Discussion: Composite outcome seem to be mortality driven since the major and minor amputations are not associated with NPS. The first paragraph might benefit some revision for the statement of composite outcome? Also in some later paragraphs.

As in the tables there are of the some more proximal endovascular procedures associated with the BTK procedures. It would be nice to include this aspect also to the discussion.

Conclusions: Nice

Author Response

Thank you very much for your insightful comments on our manuscript, “Association of the Naples Prognostic Score with Long-Term Adverse Events in Chronic Limb-Threatening Ischemia After Below-the-Knee Endovascular Revascularization.” We appreciate your positive feedback on our study and the opportunity to clarify and improve certain aspects of our manuscript. We have carefully addressed each of your suggestions, as outlined below.

Comments 1: Abstract: write out EVT and BTK. CLI right terminology?

Response 1: Abstract: We appreciate this clarification regarding the appropriate terminology and we have corrected “CLI” to “CLTI” as your suggestion. We have explained all acronyms (e.g., EVT, BTK) at their first occurrence.

Comments 2: Introduction: Only comment the above mentioned CLI/CLTI would need attention

Response 2: Introduction: We have changed to all CLI terminology to CLTI as your suggestion

Comments 3: Methods: Clopidogrel loading dose up to 600 mg? Antitrombotic treatment after first 1 month missing or not in use? ABI measured after the procedure. Not before?

Response 3: Methods: Clopidogrel was loaded between 300-600 mg according to first operator’s preferences. After the procedure, dual antiplatelet therapy was given for 1-3 months. After that, monotherapy (ASA or clopidogrel) was continued. We have added this information on methods section (line 84-86).

Thank you for your thoughtful comment regarding the ABI measurements. We recognize the importance of ABI as a measure of limb perfusion. Unfortunately, due to limitations in data collection and measurement accuracy in our study cohort, some ABI values before the procedure were missing. In follow-up, even ABI could not be measured, DUS was performed to all patients to assess the patency. However, we believe that our findings regarding NPS and overall outcomes remain robust, as our primary focus was on long-term adverse events, not directly dependent on immediate post-procedural ABI values.

Comments 4: Results: Embolectomy table 1? Embolectomy referes to an open procedure? No ABI values presented, please see general comment 

Line 191 minor instead of minör

Composite outcome seem to be mortality driven since the major and minor amputations are not associated with NPS.

Response 4: Dear reviewer thank you for noting embolectomy procedures in Table 1. In our patient cohort, thrombectomy catheter was used only selected cases with thrombus. We have revised this section as “catheter based embolectomy” to prevent confusion.

We have corrected the typographical error in line 191, changing “minör” to “minor.” (Line 191)

Thank you for your valuable insight regarding the composite outcome analysis. Our intent in creating a composite outcome was to broaden the clinical application of NPS by capturing multiple adverse events that are clinically relevant in this high-risk patient population. We aimed to include a spectrum of outcomes, such as mortality and amputation rates, to assess the predictive power of NPS in a more comprehensive manner. We recognize that mortality appears to be the primary driver of the composite outcome in our analysis, and we have clarified this aspect in the revised results and discussion section (Line 217-219, Line 324-329).

We believe that using a composite outcome enhances the potential utility of NPS as a prognostic tool, as it allows for a more holistic understanding of patient risk, encompassing both survival and major adverse limb events. We appreciate your comment and have revised the manuscript to clarify our rationale and address the composite outcome's composition in greater detail.

Comments 5: Discussion: Composite outcome seem to be mortality driven since the major and minor amputations are not associated with NPS. The first paragraph might benefit some revision for the statement of composite outcome? Also in some later paragraphs.

As in the tables there are of the some more proximal endovascular procedures associated with the BTK procedures. It would be nice to include this aspect also to the discussion.

Response 5: Dear reviewer, thank you for your valuable comments about composite outcomes. As your suggestions, we have clarified this aspect and revised Discussion section (Line 229-234, Line 324-329).

In our study, we chose to include patients, who underwent EVT for BTK arteries,  with accompanying proximal lesions because such cases reflect a common clinical scenario in which disease extends both above and below the knee. This inclusion is important to capture a more representative patient population with CLTI, who often require treatment at multiple levels to achieve optimal perfusion and clinical outcomes. And as your suggestion, we have also added some information about this aspect (Line 332-337).

Thank you once again for your constructive feedback and we hope the revised version meets the expectations of the journal.

Round 2

Reviewer 3 Report

Comments and Suggestions for Authors

Authors have revised manuscripts according given criticism. No further comments.